# Changes in Microbiota Profile in the Proximal Remnant Intestine in Infants Undergoing Surgery Requiring Enterostomy

**DOI:** 10.3390/microorganisms11102482

**Published:** 2023-10-03

**Authors:** Inês Barreiros-Mota, João R. Araújo, Cláudia Marques, Laura Sousa, Juliana Morais, Inês Castela, Ana Faria, Maria Teresa Neto, Gonçalo Cordeiro-Ferreira, Daniel Virella, Ana Pita, Luís Pereira-da-Silva, Conceição Calhau

**Affiliations:** 1Nutrition & Metabolism Department, NOVA Medical School|Faculdade de Ciências Médicas (NMS|FCM), Universidade NOVA de Lisboa, 1169-056 Lisbon, Portugal; ines.mota@nms.unl.pt (I.B.-M.); joaoricardo.araujo@nms.unl.pt (J.R.A.); claudia.sofia.marques@nms.unl.pt (C.M.); laurac_csousa@hotmail.com (L.S.); ines.castela@nms.unl.pt (I.C.); ana.faria@nms.unl.pt (A.F.); 2CHRC—Comprehensive Health Research Centre, NOVA Medical School|Faculdade de Ciências Médicas (NMS|FCM), Universidade Nova de Lisboa, 1169-056 Lisbon, Portugal; juliana.morais@nms.unl.pt (J.M.); mariateresaneto49@gmail.com (M.T.N.); l.pereira.silva@outlook.pt (L.P.-d.-S.); 3Nutrition & Metabolism Department, CINTESIS@RISE, NOVA Medical School|Faculdade de Ciências Médicas (NMS|FCM), Universidade Nova de Lisboa, 1169-056 Lisbon, Portugal; 4Neonatal Intensive Care Unit, Department of Pediatrics, Hospital Dona Estefânia, Centro Hospitalar Universitário de Lisboa Central, 1169-045 Lisbon, Portugal; goncalo.cf@chlc.min-saude.pt (G.C.-F.); danielvirella@chlc.min-saude.pt (D.V.); anamaiapita@gmail.com (A.P.); 5Medicine of Woman, Childhood and Adolescence Academic Area, NOVA Medical School|Faculdade de Ciências Médicas (NMS|FCM), 1169-056 Lisbon, Portugal

**Keywords:** enterostomy, infants, microbiota, necrotizing enterocolitis, prematurity

## Abstract

Early-life gut dysbiosis has been associated with an increased risk of inflammatory, metabolic, and immune diseases later in life. Data on gut microbiota changes in infants undergoing intestinal surgery requiring enterostomy are scarce. This prospective cohort study examined the enterostomy effluent of 29 infants who underwent intestinal surgery due to congenital malformations of the gastrointestinal tract, necrotizing enterocolitis, or spontaneous intestinal perforation. Initial effluent samples were collected immediately after surgery and final effluent samples were collected three weeks later. Gut microbiota composition was analysed using real-time PCR and 16S rRNA gene sequencing. Three weeks after surgery, an increase in total bacteria number (+21%, *p* = 0.026), a decrease in *Staphylococcus* (−21%, *p* = 0.002) and *Candida* spp. (−16%, *p* = 0.045), and an increase in *Lactobacillus* (+3%, *p* = 0.045) and in less abundant genera belonging to the *Enterobacteriales* family were found. An increase in alpha diversity (Shannon’s and Simpson’s indexes) and significant alterations in beta diversity were observed. A correlation of necrotizing enterocolitis with higher *Staphylococcus* abundance and higher alpha diversity was also observed. H2-blockers and/or proton pump inhibitor therapy were positively correlated with a higher total bacteria number. In conclusion, these results suggest that positive changes occur in the gut microbiota profile of infants three weeks after intestinal surgery.

## 1. Introduction

The establishment and evolution of the gut microbiota during early life is critical due to the mutualistic relationship that exists between the host, microorganisms, and their metabolites [1]. Multiple factors including age, mode of delivery, diet, pharmacotherapy, geographic location, and ethnicity [1,2,3,4,5] have been recognized to influence the composition of the gut microbiota in infants. Some of these factors can induce a state of dysbiosis by decreasing bacterial diversity and increasing the abundance of pathogenic bacteria [1], thereby predisposing to inflammatory, metabolic, and autoimmune diseases [6,7,8]. A typical example of dysbiosis occurs in preterm infants admitted to neonatal intensive care units (NICUs) [9]. These infants are usually nursed in incubators, receive long-term antibiotic therapy and artificial nutrition [10], and might have restricted or null breastmilk intake and scarce contact with mother’s skin [9]. In addition, it has been described that preterm infants have a gut microbiota distinct from that of full-term infants, particularly an overall lower abundance of beneficial bacteria [1]. This dysbiotic state appears to be associated with a higher risk of developing serious complications, such as sepsis and necrotising enterocolitis (NEC), which have detrimental long-term effects on infant’s health [11], including neurodevelopmental disruption [1]. 

An additional risk factor associated with gut dysbiosis in preterm infants is intestinal surgery [10]. The three main clinical conditions that usually require intestinal surgery in newborn infants are congenital malformations of the gastrointestinal tract (CMGIT), NEC, and spontaneous intestinal perforation (SIP) [12]. In these conditions, enterostomy is usually necessary; i.e., an opening into the intestine through the abdominal wall primarily used for evacuation of intestinal contents, which allows the recovery of the affected intestinal area [13,14,15]. Limited evidence exists regarding the gut microbiota composition in infants undergoing surgery requiring enterostomy [10,12,14,16]. Qian et al. [14] and Younge et al. [10] demonstrated that 1 to 3 weeks after enterostomy, the gut microbiota of newborn infants showed lower diversity, higher abundance of *Escherichia-Shigella, Klebsiella* (including potentially invasive pathogenic species)*, Streptococcus* and *Clostridium,* and low abundance of the commensal genera *Veillonella* and *Faecalibacterium*. Qian et al. [14] suggested that the enterostomy turned the gut microbiota unhealthy, although only bacteria, and not fungi, were assessed. As such, this study on infants undergoing surgery requiring enterostomy, aimed to assess changes in bacterial and fungal microbiota composition in the proximal part of the remnant intestine, and the influence of perinatal and postnatal clinical factors on the gut microbiota.

## 2. Materials and Methods

### 2.1. Study Design and Clinical Data Collection

An observational, prospective cohort study was conducted at the NICU of Hospital Dona Estefânia, Centro Hospitalar Universitário de Lisboa Central (CHULC) and NOVA Medical School, Universidade NOVA de Lisboa. Approvals were obtained from the ethical committees of CHULC (reference number 441/2017) and NOVA Medical School (reference number 50/2018/CEFCM). The study was registered at ClinicalTrials.gov (NCT 03340259), and its protocol published elsewhere [12].

The study was conducted between July 2017 and August 2020 and at that time, since no sufficient data were published that could allow an estimation of the sample size needed, a convenience, non-probabilistic, consecutive sample was recruited during the neonatal period in the NICU of Hospital Dona Estefânia, CHULC. Infants to whom an enterostomy was performed due to CMGIT, NEC, or SIP diagnosis were eligible, and were included for analysis if they remained in the NICU for a period of at least 21 days after surgery. The study period was based on the average length of stay of infants undergoing enterostomy due to the studied conditions in our NICU [12]. 

Written informed consent was obtained from legal representatives of infants prior to their inclusion in the study.

Detailed clinical data (sociodemographic, perinatal, and postnatal parameters) from infants were retrieved from medical records [12]. Gestational age is expressed in completed weeks and postnatal age in completed days. Regarding the indications to perform enterostomy, recorded surgical diagnoses were considered.

### 2.2. Enterostomy Effluent Samples Collection and DNA Extraction

Enterostomy effluent samples were collected when at least 2 mL were available, and sampling was performed every 3 days up to 21 days after the first collection. Bacterial and fungal DNA were extracted and purified from samples using an NZY Tissue gDNA Isolation Kit (NZYTech, Lisbon, Portugal), as previously described [12].

### 2.3. Quantitative Analysis of Faecal Microbiota by RT-PCR

Total bacteria and *Candida* numbers were quantified by quantitative real-time PCR (RT-PCR) using LightCycler^®^ (Roche Applied Science, Indianapolis, ID, USA) as previously described [12]. Primer sequences used to target bacterial 16S rRNA genes were described elsewhere [12]. Results are expressed as log_10_ 16S rRNA gene copies/ng of DNA. 

### 2.4. Microbial 16S rRNA Sequence Analysis

Samples of the enterostomy effluent were analysed by 16S rRNA sequencing. Libraries were processed and sequenced following the 16S Metagenomic Sequencing Library Preparation protocol from Illumina (Illumina; San Diego, CA, USA). Primers used to capture the region V3–V4 of the bacterial 16S region (primers 515F: GTGYCAGCMGCCGCGGTAA; 806R: GGACTACNVGGGTWTCTAAT) were previously described by Walters and colleagues [17]. The samples were pooled and loaded into the Illumina MiSeq System and sequenced using a 280-multiplex approach on a 2 × 250 bp run, according to manufacturer’s procedures [17]. For all sequencing reads, QIIME 2.11 was used with default parameters for demultiplexing, quality filtering, and clustering reads into operational taxonomic units—OTUs. The Greengenes database was used for taxonomy assignment [18].

### 2.5. Bioinformatic Analysis

The microbiota data analysis was performed using Microbiome Analyst, a web-based tool for comprehensive statistical, visual, and meta-analysis of microbiome data [19]. Gut microbial richness (total number of species) was measured by Chao1 [20]. Alpha diversity was measured by both Shannon and Simpson indexes, which summarize the species richness and evenness (abundance distribution across species) within a sample [9]. The Shannon index is more sensitive to the richness of the community, while Simpson’s index puts more weight on species evenness [9,20]. Beta diversity was used to create the principal coordinates analysis (PCoA) plot based upon Bray–Curtis dissimilarity, to evaluate differences in the community of bacterial species according to the experimental factor time (initial and final). Distances (or dissimilarities) between samples of the same group were compared to the distances between groups using PERMANOVA [19].

### 2.6. Statistical Analysis

Statistical analysis was performed using SPSS^®^ software version 25 (IBM SPSS Statistics Corporation, Chicago, IL, USA) and the GraphPad Prism^®^ version 8 (GraphPad Software^®^, San Diego, California, USA). Data normality was assessed using Shapiro–Wilk, D’Agostino–Pearson omnibus, and Kolmogorov–Smirnov tests. For normally distributed variables, data are presented as mean ± standard deviation (SD), and paired Student’s *t*-test was used to compare changes from the baseline. For non-normally distributed variables, data are presented as median [interquartile range], and the Wilcoxon test was used to compare changes from the baseline. Correlation analysis between the variation in microbiota parameters and clinical data was performed using Pearson correlation coefficients for normally distributed continuous variables, and point-biserial correlation coefficients for binary categorical variables. The absolute magnitude of the correlation coefficient (rs) was interpreted as negligible (0.00–0.10), weak (0.10–0.39), moderate (0.40–0.69), strong (0.70–0.89), or very strong (0.90–1.00) [21]. Differences were considered to be statistically significant when *p* < 0.05. 

## 3. Results

### 3.1. Clinical and Demographic Characteristics of Infants

From a total of 32 eligible infants, 3 were excluded due to the limited amount of enterostomy effluent available for microbiota analysis. Therefore, 29 infants were included in the study. Clinical and demographic characteristics of included infants are shown in Table 1. There was no significant predominance of sex (males 59%) or mode of delivery (caesarean section 55%). The median (IQR) gestational age was 33 weeks (27–37) and the median birthweight was 1779 g (885–2485). Clinical conditions were NEC in 13 (45%), CMGIT in 12 (41%), and SIP in 4 (14%) infants. In six infants, data on enteral feeding prior to surgery were missing; from the remaining 23 infants, 15 were enterally fed prior to surgery. The median (IQR) age of infants at time of enterostomy was 2 days of life (1–8). Twenty-six (90%) enterostomies were performed at the small intestinal level (twenty at the ileal level and six at the jejunal level) and three (10%) at the colonic level. On the 21st day of the study follow-up, the median (IQR) weight gain of infants was 332 g (192–466). During the study period, 26 (90%) infants received antibiotic therapy, 9 (31%) antifungal therapy, and 8 (28%) H_2_-receptor antagonists. Sepsis was confirmed by blood culture in 11 (38%) infants. During at least 14 of the 21-day study period, 14 infants (48%) were fed human milk, 13 (45%) were fed formula, and 2 (7%) were on exclusive parenteral nutrition. 

### 3.2. Microbiota Profile of Enterostomy Effluents

Deviations from the study protocol [12] due to an unexpected lack of funding are acknowledged. First, the microbiota profile analysis of enterostomy effluents was limited to the initial sample and to the final sample. Second, it was only possible to analyse the number of total bacteria and *Candida* spp. by qPCR (*Staphylococcus*, *Bifidobacterium*, *Bacteroides fragilis*, and *Escherichia coli* not included), thus priority was given to 16S rRNA gene analysis.

During the study period, the number of total bacteria increased by 21% (Figure 1A) and the number of *Candida* spp. decreased by 16% (Figure 1B). Due to the limited volume of final samples, a subset of 23 infants, instead of 29, was considered for 16S rRNA sequencing analysis. *Proteobacteria*, *Firmicutes*, *Bacteroidetes,* and *Actinobacteria* were the dominant phyla, but no significant changes in the median relative abundances between initial and final samples were observed: *Proteobacteria* increased from 48% to 68%, *Firmicutes* increased from 46% to 26%, *Bacteroidetes* was 4% in both, and *Actinobacteria* decreased from 2% to 1% (Figure 2A). Also, no significant differences were observed in the relative abundance of other phyla.

The most abundant bacterial genera in initial samples were *Staphylococcus* (36%), *Haemophilus* (5%), and *Streptococcus* (5%), whereas in final samples were *Streptococcus* (11%), *Staphylococcus* (5%), *Lactobacillus* (3%), and *Enterococcus* (3%) (Figure 2B). Among these genera, the relative abundance of *Staphylococcus* decreased (from 35.453% to 4.662%, *p* = 0.002) and the relative abundance of *Lactobacillus* increased (from 0.054% to 3.263%, *p* = 0.045) during the study period. No significant changes were observed in the other prevalent genera. Significant changes in the relative abundance of low prevalent genera were also observed during the study period, such as an increase in *Klebsiella* (from 0.005% to 0.011%, *p* = 0.029) and *Citrobacter* (from 0.013% to 0.064%, *p* = 0.002) and a decrease in *Corynebacterium* (from 2.118% to 0.033%, *p* = 0.021), *Bacillus* (from 0.006% to 0.001%, *p* = 0.010), *Jeotgalicoccus* (from 0.001% to 0.000%, *p* = 0.016), *Sphingomonas* 0.014% to 0.000%, *p* = 0.008), and *Turicibacter* (0.000% to 0.005%, *p* = 0.032).

From the beginning to the end of the study period, a high variability among participants at the individual level was observed in the relative abundance of the most prevalent bacterial phyla (Figure 3A) and genera (Figure 3B). 

At the level of alpha diversity (i.e., abundance of species heterogeneity), both Shannon’s and Simpson’s indexes increased in the final samples, while the Chao1 index remained unaltered (Figure 4A–C). At the level of beta diversity (i.e., unsimilarity of microbiota communities at the species level), significant differences (*p* < 0.023) were observed among the microbiota communities during the study period (Figure 5). 

### 3.3. Correlations between Microbiota and Clinical Parameters

To determine if changes in microbiota profile between the initial and the final samples were associated with clinical parameters, Pearson’s correlation analysis was performed between the variation in microbiota parameters (i.e., difference between day 21 and 1) and clinical parameters. No significant correlations were found between the variation in total bacteria number, the relative abundance of genera or alpha-diversity indexes and most clinical parameters. The only exceptions were a positive and moderate correlation between NEC and increases in *Staphylococcus* abundance, Shannon’s index, or Simpson’s index (Table 2). Moreover, post-surgical exposure to H_2_-blockers and/or proton pump inhibitor therapy was positively and weakly correlated with an increase in total bacteria number (Table 2). 

## 4. Discussion

This cohort study performed in infants subjected to gastrointestinal surgery requiring enterostomy, aimed to investigate the development of gut microbiota composition three weeks after surgery. Although no significant changes were observed in bacterial phyla, apparent beneficial changes were found in bacteria genera. These included an increase in the total number of bacteria, a decrease in *Staphylococcus* and *Candida* spp., and an increase in *Lactobacillus*. A significant increase in alpha diversity and significant differences in beta diversity were also observed. A correlation was found between NEC diagnosis and both a higher abundance of *Staphylococcus* and alpha diversity, and between post-surgical exposure to H_2_-blockers and/or proton pump inhibitors and a higher number of total bacteria. 

Changes in gut microbiota in early life have a long-lasting impact on the development and establishment of the gut microbiota throughout life [10,22]. Moreover, early-life gut dysbiosis has been associated with an increased risk of inflammatory, metabolic, and immune diseases later in life [23]. In this study, we took advantage of infants with intestinal surgery having an enterostomy, allowing the direct collection of fresh intestinal effluent [15]. As the effluent was collected from the proximal enterostomy, we assumed that the analysed microbiota represented the one located closest to the stoma [12,24,25], also reflecting the colonisation in all proximal remnant parts of the intestine. 

Knowledge about the effects of intestinal surgery on changes in gut microbiota composition is scarce. One important factor affecting the gut microbiota after surgery may be the removal of a large number of microorganisms along with the resected or excluded intestine. In this study, the 16S rRNA sequence analysis was used to identify the whole microbiota [26]. The targeted quantitative PCR with specific primers to genus/species allowed the quantification of microbiota [27]. The 18S rRNA gene analysis was also used to examine fungi [28].

From the beginning until the end of the three-week study period, we observed a 20% increase in the total number of bacteria after surgery. Moreover, diversity indexes revealed a change in the intestinal microbiota community towards a higher number of species and their relative abundance, from the beginning to the end of the study period. These results are in contrast to those reported by Qian et al. [14] and Younge et al. [10], who found a lower Shannon’s diversity index in the enterostomy fluid, although the total number of bacteria was not evaluated. This discrepancy may be related to different underlying clinical conditions studied, since in the study by Qian et al. [14], 27% of samples were cases of Hirschprung disease, and in the study by Younge et al. [10], practically no cases of GMTI were included, whereas our study included 41% of CMGIT and none of Hirschprung disease. Another explanation may be related to the different types of infant feeding, since in our study the percentage of newborns fed human milk was higher than those in the Qian et al. [14] and Younge et al. [10] studies (48% vs. 27% vs. 38%, respectively).

Regarding the abundance of bacterial genera, in our study, a decrease in *Staphylococcus* and an increase in *Lactobacillus* was found. *Staphylococcus* is highly prevalent in the gut of preterm infants during the first postnatal week [1,11,23], and may include either opportunistic pathogen species (such as *S. aureus*) [29] or commensal species (such as *S. epidermidis)* [30]. On the other hand, *Lactobacillus* are commensal bacteria characteristically found in the small intestine of both term and preterm infants [8,14]. The genus *Lactobacillus* can promote the development of infant-acquired and innate immunity in early life [2] and its abundance has been associated with positive effects on the intestinal epithelium, in particular, the maintenance of its integrity [8,14]. In addition, some strains of Lactobacillus are able to partially consume relevant human milk oligosaccharides [30]. Despite the low prevalence of *Klebsiella* and *Citrobacter*, an increase in their abundance was observed at the end of our study period. Both genera belong to the *Enterobacterales* family, whose abundance in preterm infants with enterostomy was reported to be correlated with inflammatory-associated conditions [10], such as NEC and sepsis [29]. In fact, this association is expected since the *Enterobacterales* family includes several pathogenic *Klebsiella* and *Citrobacter* species. However, in our study, the abundance of *Klebsiella* or *Citrobacter* was not correlated with the prevalence of NEC or sepsis, probably due to the low relative abundance (<0.07%) of these genera detected after enterostomy. 

Concerning fungi genera, we observed that the abundance of *Candida*, a fungal taxon frequently present in the preterm infants’ gut mycobiome [11], decreased three weeks after enterostomy. To the best of our knowledge, this is the first time that a short-term change in *Candida* abundance has been reported. *Candida* is considered a pathobiont, i.e., under a host healthy state, it acts as a commensal, but under certain host conditions (e.g., loss of intestinal epithelial integrity), it can cause or promote disease [11]. 

Overall, our results suggest that changes in gut microbiota profile in the proximal remnant part of the intestine, shortly after intestinal surgery, appear not to be deleterious for the infants’ health, in contrast to gut microbiota profiles reported in other similar studies [10,14]. Our findings should however be interpreted with caution since no long-term clinical and microbiological follow-up was carried out. 

Since several factors could modulate the composition of gut microbiota in the studied individuals, including age and birthweight [1,2], mode of delivery [1,2], underlying surgical condition [14], level of enterostomy [14], type of feeding [1,2,5,10,14], and pharmacotherapy [1,2,5,31], a correlation analysis between these factors and gut microbiota data was performed. Despite the lack of correlation between most of these clinical factors and the total number of bacteria, the relative abundance of bacterial and fungi genera, or the bacteria diversity indexes of these clinical factors, a few significant associations were observed. At the end of the study, a greater abundance of *Staphylococcus* and a greater bacterial diversity were associated with previous NEC. Fu et al. [32] reported that before NEC development, intestinal microbial richness and evenness were higher than in those who did not develop NEC. However, after treatment, in infants who developed NEC, the microbial diversity did not change significantly over time and the bacteria at the top of each taxonomic level were similar to infants without a diagnosis of NEC [33]. These authors suggested that this association may have resulted from the high inter-infant variability of the sequencing data obtained, also observed in our study. We also found an association between a higher total number of bacteria and H_2_- blockers and/or proton pump inhibitors therapy after surgery. H_2_-blockers and/or proton pump inhibitors reduce the acidity in the stomach, allowing the growth of acid-sensitive bacteria, in particular, those belonging to the *Proteobacteria* phylum [32,34]. A study in preterm infants found an increased abundance of *Proteobacteria* in infants receiving H_2_-blockers compared to those who did not [31]. In this context, *Proteobacteria* may have contributed to the positive association between the total number of bacteria and H_2_ blocker therapy found in this study. 

To the best of our knowledge, this is the first observational, longitudinal, prospective study in infants assessing the colonisation of the proximal remnant part of the intestine, shortly after intestinal surgery. Another strength of this study is the gene analysis to examine fungi abundance, seldom included in studies characterising neonatal intestinal microbiota [10,12,24,34,35,36].

Some limitations need, however, to be acknowledged. First, although this is a single-centre study, our unit is a tertiary referral centre for neonatal surgery and receives infants from several other neonatal units, thereby allowing a certain degree of representability of results obtained from a large population of infants. Second, a convenience, non-probabilistic sample was used since there were insufficient published data to calculate the sample size. To mitigate biased data related to convenience sampling, a systematic and consecutive sample during a three-year period was used. Finally, the short three-week follow-up period, besides the sample size, may explain the lack of significant associations between most clinical factors and microbiota data.

## 5. Conclusions

This study demonstrates that the gut microbiota profile of preterm infants changes three weeks after intestinal surgery. These changes include an increase in the total abundance and diversity of bacteria, a decrease in *Staphylococcus* and *Candida,* and an increase in *Lactobacillus* and in low abundant genera belonging to the *Enterobacteriales* family. In general, these changes do not appear to be deleterious, and may even be beneficial for the gut health of infants, due to a decrease in opportunistic microbial species and an increased abundance of commensals, contributing to a greater richness and diversity. However, more powered and long-term studies are needed to confirm these findings. A better understanding of how changes in the gut microbiota affect clinical outcomes may increase knowledge towards reducing morbidity and improving health outcomes in this population of high-risk infants.

## Figures and Tables

**Figure 1 microorganisms-11-02482-f001:**
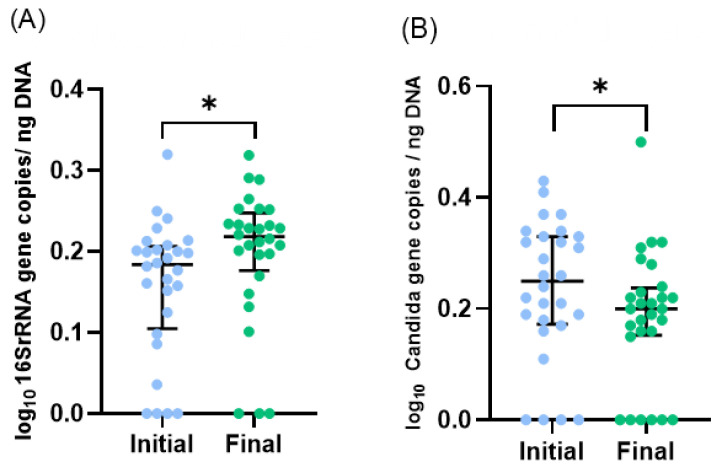
Microbiota profile of initial and final effluents samples (*n* = 28, one sample was excluded as an outlier): (**A**) Total bacteria and (**B**) *Candida* spp. numbers were assessed by qPCR. Scatter dot plot with median [interquartile range]. * *p* < 0.05 (Wilcoxon test).

**Figure 2 microorganisms-11-02482-f002:**
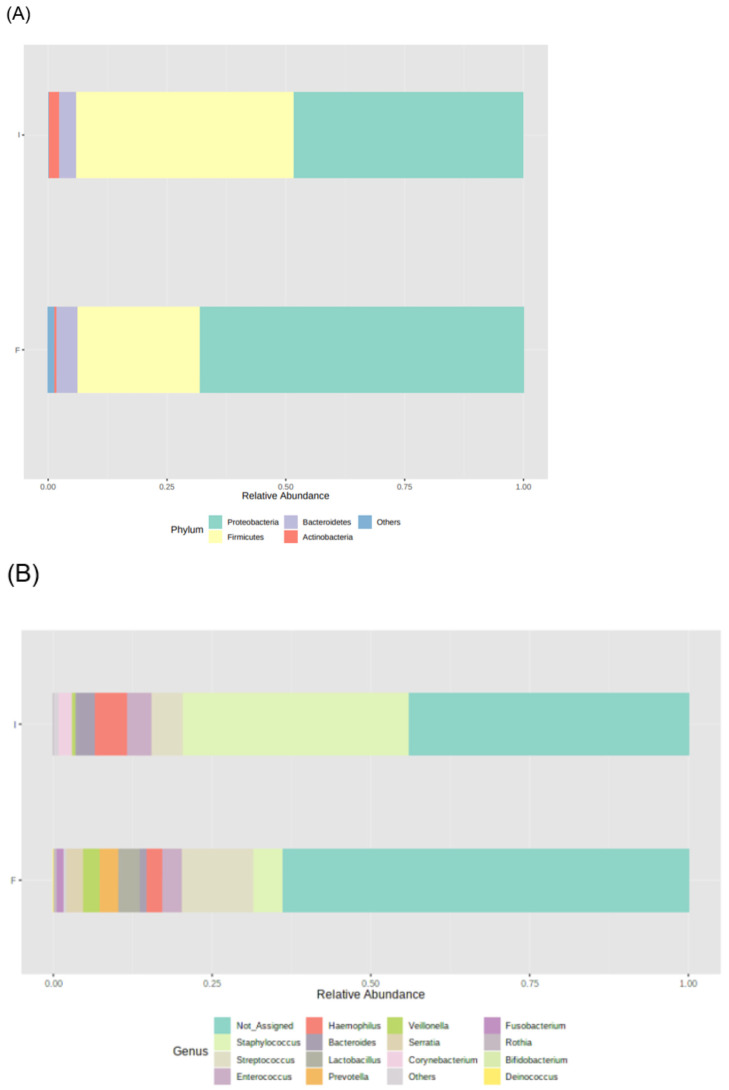
Microbiota profile of initial (I) and final (F) effluents samples (*n* = 23): Bacterial phyla (**A**) and genera (**B**) relative abundance, using 16S rRNA gene analysis. Each taxon is represented by a different colour.

**Figure 3 microorganisms-11-02482-f003:**
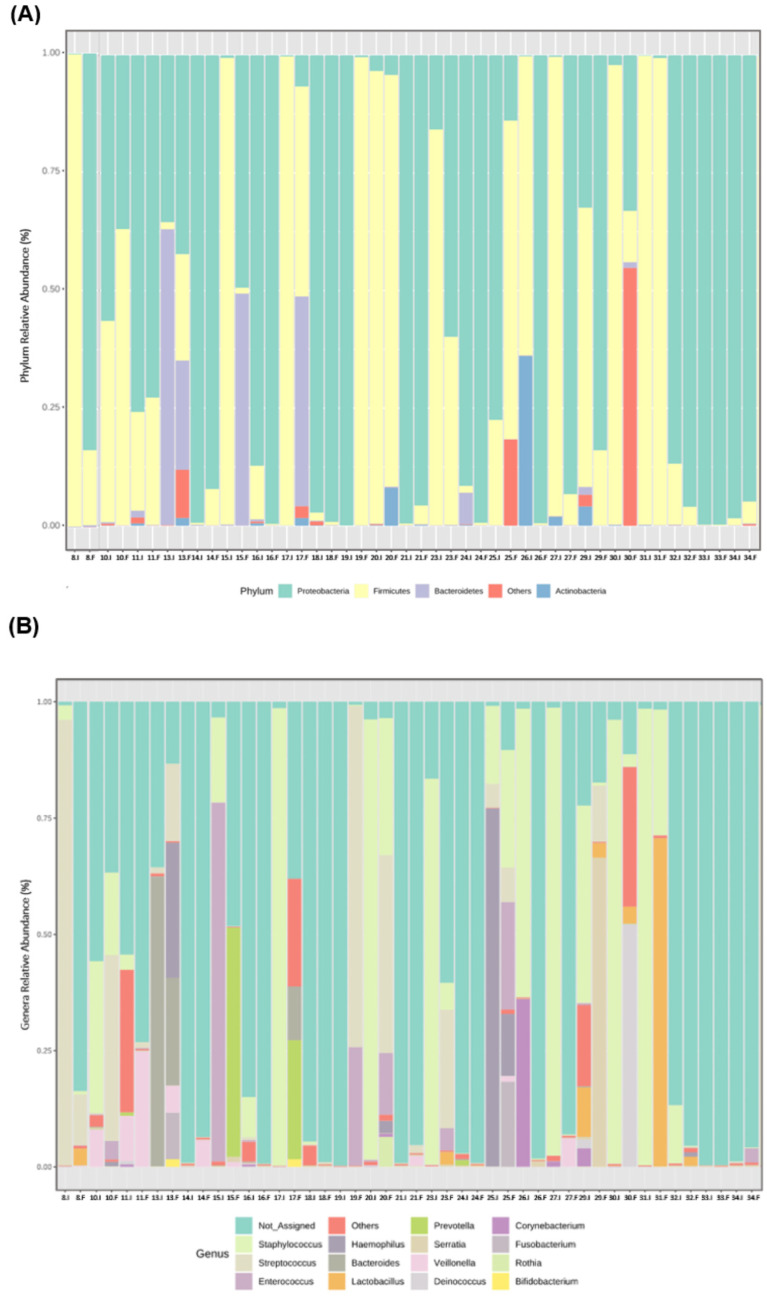
Individual microbiota profile of initial (I) and final (F) effluents samples (*n* = 23): Bacterial phyla (**A**) and genera (**B**) relative abundance, using 16S rRNA gene analysis. Each taxon is represented by a different colour. Two consecutive bars represent initial and final samples for each participant.

**Figure 4 microorganisms-11-02482-f004:**
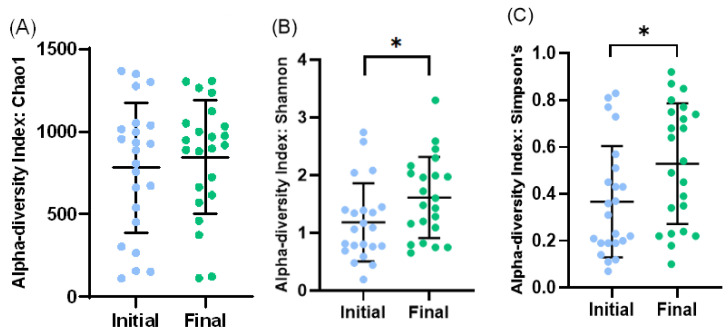
Bacterial diversity in initial and final effluent samples (*n* = 23): Species richness measured by Chao1 (**A**) index. Alpha diversity measured by Shannon’s diversity index (**B**) and Simpson’s index (**C**). Scatter dot plot with mean and standard deviation. * *p* < 0.05 (*t*-test).

**Figure 5 microorganisms-11-02482-f005:**
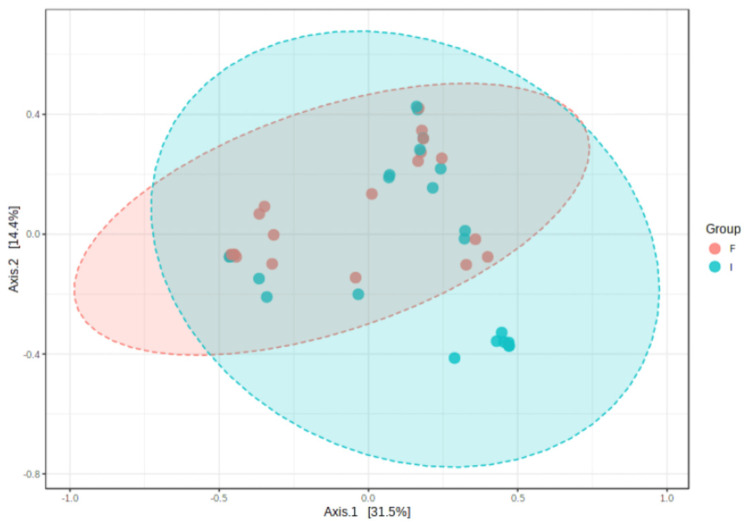
Clustering of bacterial genera communities by principal coordinates analysis (PCoA) (*n* = 23). Data are plotted according to the first two components, which explain 31.5% (PCo1) and 14.4% (PCo2) of intestinal effluent microbiota variation between the initial (I) and final (F) samples. Each point represents one sample, coloured according to the time point of sample collection.

**Table 1 microorganisms-11-02482-t001:** Clinical and demographic parameters of infants.

Parameter	
Male sex, *n* (%)	17 (59%)
C-section delivery, *n* (%)	16 (55%)
Gestational age, weeks, median (IQR)	33 (27–37)
Birth weight, g, median (IQR)	1779 (884–2485)
Apgar score at 5 min, median (IQR)	9 (8–10)
Peripartum antibiotics, *n* (%)	6 (23%) *
Surgical condition - CMGIT, *n* (%) - NEC, *n* (%) - SIP, *n* (%)	12 (41%) 13 (45%) 4 (13.8%)
Level of enterostomy - Small intestine **, *n* (%) - Colon, *n* (%)	26 (90%) 3 (10%)
Age at enterostomy, postnatal days, median (IQR)	2 (1–8)
Body weight gain during the study period, g, median (IQR)	332 (192–466)
Antibiotic therapy, *n* (%)	26 (90%)
Antifungal therapy, *n* (%)	9 (31%)
H_2_-blockers and/or proton pump inhibitors, *n* (%)	8 (28%)
Sepsis confirmed by blood culture, *n* (%)	11 (38%)
Type of nutrition post enterostomy - Human milk, *n* (%) - Formula, *n* (%) - Exclusive parenteral nutrition, *n* (%)	14 (48%) 13 (45%) 2 (7%)

Data are expressed as median [interquartile range] for continuous variables without normal distribution and as number (percentage) for categorical variables. * 3 infants were excluded since information was missing in their clinical records. ** 20 at ileal level and 6 at jejunal level. CMGIT—congenital malformations of the gastrointestinal tract; IQR—interquartile range; NEC—necrotizing enterocolitis; SIP—spontaneous intestinal perforation.

**Table 2 microorganisms-11-02482-t002:** Identified clinical parameters associated with changes in microbiota profile over time (Pearson correlation coefficient).

	*Staphylococcus*	Shannon’s Index	Simpson’s Index	Total Bacteria Number
NEC	−0.436 *	0.468 *	0.483 *	0.004
H_2_-blockers and/or proton pump inhibitors	0.043	0.198	0.200	0.385 *

Correlations between dichotomous categorical variables and continuous variables were performed using point biserial correlation coefficient. * *p* < 0.05. NEC—necrotizing enterocolitis.

## Data Availability

Deidentified individual participant data supporting the findings of this study are indefinitely available within the article or the following dataset (DOI: 10.17632/x4v75f7bbp.1).

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
