# Peer review of "Changes in Microbiota Profile in the Proximal Remnant Intestine in Infants Undergoing Surgery Requiring Enterostomy"

_microorganisms, 2023, doi:10.3390/microorganisms11102482_

Round 1

Reviewer 1 Report

microorganisms
Article
Changes in microbiota profile in the proximal remnant intestine in infants undergoing surgery requiring enterostomy

In this manuscript, authors gave a correlation of necrotizing
enterocolitis with higher Staphylococcus abundance and higher alpha-diversity was also observed.
H2-blockers and/or proton pump inhibitor therapy were positively correlated with a higher total
bacteria amount, results suggesting positive changes in gut microbiota profile three weeks after
intestinal surgery in infants.

The article was well-written and well-organized.
The authors have reviewed the current state of knowledge
In conclusion, this study demonstrates that the gut microbiota profile of preterm infants changes 21 days after intestinal surgery. These changes include an increase in total bacteria abundance and diversity, a decrease in Staphylococcus and Candida, and an in-
crease in Lactobacillus and in low abundant genera belonging to the Enterobacteriales family.
Add heading Conclusions

Figures (5) and tables (2) are adequate, especially some of the figures are missing clarity, please change,
The literature was up to the mark. Covering most of the required articles.

Plea check for GRAMMAR,
The article is well-suitable and can be accepted for publication in microorganisms after minor revisions.

Author Response

Changes in microbiota profile in the proximal remnant intestine in infants undergoing surgery requiring enterostomy.

In this manuscript, authors gave a correlation of necrotizing enterocolitis with higher Staphylococcus abundance and higher alpha-diversity was also observed.
H2-blockers and/or proton pump inhibitor therapy were positively correlated with a higher total bacteria amount, results suggesting positive changes in gut microbiota profile three weeks after intestinal surgery in infants.

The article was well-written and well-organized.

The authors have reviewed the current state of knowledge.

In conclusion, this study demonstrates that the gut microbiota profile of preterm infants changes 21 days after intestinal surgery. These changes include an increase in total bacteria abundance and diversity, a decrease in Staphylococcus and Candida, and an increase in Lactobacillus and in low abundant genera belonging to the Enterobacteriales family.

The authors are very grateful for the insightful comments and suggestions that helped to improve the manuscript. A point-to-point response to questions is provided and changes are highlighted in the revised manuscript in blue font.

Add heading Conclusions.

Response: Conclusions heading was added as suggested.

Figures (5) and tables (2) are adequate, especially some of the figures are missing clarity, please change.

Response: Thank you for the comment. Figures and graphics were assembled from the original figures automatically generated by the software ‘Microbiome Analyst’ described in Methods. Therefore, the original formatting cannot be changed. However, the resolution quality of figures was improved, and tables were reformatted.

The literature was up to the mark, covering most of the required articles.

Response: Thank you for the comment.

Please check for GRAMMAR.

Response: thank you for the suggestion. A careful grammar and wording revision of the English was performed by a native English speaker.

The article is well-suitable and can be accepted for publication in microorganisms after minor revisions.

Response: Thank you for the comment.

Reviewer 2 Report

Manuscript ID: microorganisms-2649447

Title: Changes in microbiota profile in the proximal remnant intestine in infants undergoing surgery requiring enterostomy

 The manuscript may be published in this journal, but the author needs to carefully revise the following aspects:

(1) Lines must be numbered. The lack of line numbering makes it difficult to nest and target the comments of the Reviewer.

(2) The overall English needs to be improved. Please seek guidance from a native English speaker if possible ("the" "a", commas, plural form, and others could be corrected).

(3) References used should be updated, where it is noted (no reference in 2023 and one reference in 2022)

(4) In each page: " Microorganisms 202210, x FOR PEER REVIEW ", update to 2023

(5) Figures (2,3 and 5) need to improve the resolution quality

(6) In Abstract: Change " collected " to "was collected "

 (7) At the end of the Abstract, a conclusion of the manuscript should be written.

 (8) In introduction and references list: Please check " Quian et al.[10] " or " Qian et al.[10] " is correct

(9) In introduction: Change "might have restricted" to "and might have restricted"

(10) In 2.2. Enterostomy Effluent Samples Collection and DNA extraction: Change "performed" to "was performed"

(11) In 2.6. Statistical Analysis: Change "ttest" to "t-test"

(12) Change "Table 2. Identified clinical parameters associated to " to "Table 2. Identified clinical parameters associated with"

(13) Change " may be related with" to " may be related to"

(14) Change " positive effects in " to " positive effects on"

(15) Change " three year-period" to " three-year period"

(16) The Discussion section needs serious improvements. Authors have to focus on discussing their own findings and interpreting these results.

(17) Please follow the authors' instructions in writing the reference in the list. For references about textbooks, please add the page numbers of the textbook. Also, please add the city of the publisher.

Extensive editing of English language required

Author Response

Title: Changes in microbiota profile in the proximal remnant intestine in infants undergoing surgery requiring enterostomy.

The manuscript may be published in this journal, but the author needs to carefully revise the following aspects:

The authors are very grateful for the insightful comments and suggestions that helped to improve the manuscript. A point-to-point response to questions is provided and changes are highlighted in the revised manuscript in blue font.

(1) Lines must be numbered. The lack of line numbering makes it difficult to nest and target the comments of the Reviewer.

Response: as suggested, lines were numbered.

(2) The overall English needs to be improved. Please seek guidance from a native English speaker if possible ("the" "a", commas, plural form, and others could be corrected).

Response: thank you for the comment and suggestion. A careful grammar and wording revision of the English was performed by a native English speaker.

(3) References used should be updated, where it is noted (no reference in 2023 and one reference in 2022)

Response: thank you for the comment. In fact, the literature related to the topic of the manuscript (microbiota profile in infants with enterostomy) is very scarce. We did not find specific articles on this topic published in the year 2023. However, to comply with the suggestion, we added more recent articles published in 2022 (references 3,4, and 5), directly or indirectly related to the topic.

(4) In each page: " Microorganisms 2022, 10, x FOR PEER REVIEW ", update to 2023

Response: thank you. The correction was made in accordance.

(5) Figures (2,3 and 5) need to improve the resolution quality.

Response: Figures 2, 3 and 5 were assembled from the original figures automatically generated by the software ‘Microbiome Analyst’ described in Methods. Therefore, the original formatting cannot be changed. However, the resolution quality of these figures was improved.

(6) In Abstract: Change " collected " to "was collected "

Response: thank you very much. The change was made in accordance.

 (7) At the end of the Abstract, a conclusion of the manuscript should be written.

Response: a conclusion was included at the end of the Abstract, as suggested.

(8) In introduction and references list: Please check " Quian et al.[10] " or " Qian et al.[10] " is correct

Response: thank you very much for the correction.

(9) In introduction: Change "might have restricted" to "and might have restricted"

Response: thank you very much. The change was made in accordance.

(10) In 2.2. Enterostomy Effluent Samples Collection and DNA extraction: Change "performed" to "was performed".

Response: thank you very much. The change was made in accordance.

(11) In 2.6. Statistical Analysis: Change "ttest" to "t-test"

Response: thank you very much for the correction.

(12) Change "Table 2. Identified clinical parameters associated to "to" Table 2. Identified clinical parameters associated with"

Response: thank you very much for the correction.

(13) Change "may be related with" to "may be related to"

Response: thank you very much. The change was made in accordance.

(14) Change "positive effects in "to" positive effects on"

Response: thank you very much for the correction. The change was made in accordance.

(15) Change " three year-period" to " three-year period"

Response: thank you very much. The change was made in accordance.

(16) The Discussion section needs serious improvements. Authors have to focus on discussing their own findings and interpreting these results.

Response: as suggested, the Discussion section was improved by discussing and interpreting our findings in more detail.

(17) Please follow the authors' instructions in writing the reference in the list. For references about textbooks, please add the page numbers of the textbook. Also, please add the city of the publisher.

Response: Some references were misquoted and may have been confused with textbooks. They have been corrected. There are no books included in the reference list.